# Applying Machine Learning for Enhanced MicroRNA Analysis: A Companion Risk Tool for Oral Squamous Cell Carcinoma in Standard Care Incisional Biopsy

**DOI:** 10.3390/biom14040458

**Published:** 2024-04-09

**Authors:** Neha Pruthi, Tami Yap, Caroline Moore, Nicola Cirillo, Michael J. McCullough

**Affiliations:** Melbourne Dental School, The University of Melbourne, Carlton, VIC 3053, Australia; npruthi@student.unimelb.edu.au (N.P.); tspyap@unimelb.edu.au (T.Y.); moore@unimelb.edu.au (C.M.); nicola.cirillo@unimelb.edu.au (N.C.)

**Keywords:** oral squamous cell carcinoma, oral potentially malignant disorders, early detection, machine learning, lichen planus, leukoplakia, artificial intelligence, biomarker

## Abstract

Machine learning analyses within the realm of oral cancer outcomes are relatively underexplored compared to other cancer types. This study aimed to assess the performance of machine learning algorithms in identifying oral cancer patients, utilizing microRNA expression data. In this study, we implemented this approach using a panel of oral cancer-associated microRNAs sourced from standard incisional biopsy specimens to identify cases of oral squamous cell carcinomas (OSCC). For the model development process, we used a dataset comprising 30 OSCC and 30 histologically normal epithelium (HNE) cases. We initially trained a logistic regression prediction model using 70 percent of the dataset, while reserving the remaining 30 percent for testing. Subsequently, the model underwent hyperparameter tuning resulting in enhanced performance metrics. The hyperparameter-tuned model exhibited high accuracy (0.894) and ROC AUC (0.898) in predicting OSCC. Testing the model on cases of potentially malignant disorders (OPMDs) revealed that leukoplakia with mild dysplasia was predicted as having a high risk of progressing to OSCC, emphasizing machine learning’s advantage over histopathology in detecting early molecular changes. These findings underscore the necessity for further refinement, incorporating a broader set of variables to enhance the model’s predictive capabilities in assessing the risk of oral potentially malignant disorders.

## 1. Introduction

The term oral potentially malignant disorders refers to any type of mucosal abnormality associated with an increased risk of developing oral squamous cell carcinoma (OSCC) [1]. Among these conditions are oral leukoplakia, oral lichen planus (OLP), oral submucous fibrosis (OSF), oral lichenoid lesions and oral erythroleukoplakia (OEL) [1]. According to Iocca [2], carcinogenesis rates across all types of OPMDs are 7.9%. In the individual OPMD groups, cancer rates range from 1.4% for OLPs, 9.5% for leucoplakias, 3.8% for OLLs, to 5.2% for oral submucous fibrosis. Patients with OSCC have an overall survival rate of less than 60% after 5 years, which adversely affects their mental health and quality of life [3]. Hence, early detection is the key.

The natural history of OSCC is not fully understood; some OPMDs progress to transform into malignant tumours, some remain stable, and others return to health [4]. Moreover, OSCC can develop from lesions that have no history of epithelial dysplasia [5] or from mucosa that appears normal but contains significant molecular abnormalities [6,7]. In the past decades, the effectiveness of OSCC detection has not much improved [8].

While histopathology is widely regarded as the gold standard for cancer detection, it is not without its drawbacks. One significant limitation lies in its inability to distinguish between dysplastic and nondysplastic lesions with any reliable precision. Subtle variations in epithelial changes may pose challenges, potentially leading to misinterpretation or oversight of lesions carrying a risk of malignancy. Moreover, the process of obtaining tissue samples through biopsy is invasive and may not always capture the entirety of the lesion, resulting in sampling errors. Additionally, histopathological evaluations are dependent on the expertise of the pathologist, and variations in interpretation may occur. There is a lack of awareness among clinicians and sometimes they may also miss clinical signs [1]. These drawbacks highlight the need for supplementary diagnostic tools that can address these limitations, offering a more comprehensive and accurate approach to cancer detection. It is crucial to identify OSCC at a molecular level before there are evident clinical and histologic changes [9].

MicroRNAs (miRNAs) have emerged as markers in the field of cancer, significantly contributing to our understanding of the molecular aspects of the disease. miRNAs play a role in regulating genes after they are transcribed, influencing various cellular processes [10]. In oncogenesis, microRNAs contribute to the modulation of key oncogenic and tumour suppressor proteins [11]. Exploring miRNAs as markers not only has the potential to enhance early detection but also opens up possibilities for tailored therapeutic options that aim to improve overall prognosis for those affected by this condition. The markers selected for our study encompass a carefully chosen set of microRNAs, including miR-21, miR-100, let-7c, miR-24, miR-99, and miR-125b, which have been shown in the literature to be dysregulated in OSCC [9,12,13,14,15,16,17].

Machine learning, a subset of artificial intelligence (AI), empowers computers to learn from data and make predictions [18]. Within the realm of machine learning, logistic regression stands out as a statistical model utilizing the logistic function to model binary dependent variables [19]. Logistic regression is a form of machine learning employed in various fields, including oncology, where it proves valuable for predicting diagnoses, recurrence, metastasis, and prognosis [20]. In this study, we utilized logistic regression, showcasing its efficacy as a type of machine learning technique in our investigation of oral squamous cell carcinoma risk prediction. 

The integration of histopathological insights with PCR assessment of microRNA expression from FFPE samples serves as a powerful tool for advancing cancer prediction and diagnosis. By combining the strengths of tissue morphology and molecular signatures, this integrated approach provides a comprehensive understanding of underlying biological mechanisms. Moreover, the incorporation of machine learning, specifically logistic model analysis, transforms this combination into an advanced diagnostic tool, enhancing predictive accuracy and diagnostic precision. This tool offers a promising avenue for researchers and clinicians, providing a robust framework to overcome the limitations associated with individual methods and significantly improving the effectiveness of cancer diagnostics. Such a tool could ensure appropriate treatments for patients and potentially prevent the progression of these disorders into malignancy.

The primary objective of this study involves the application of machine learning models to assess the risk of OSCC based on microRNA analysis from standard care incisional biopsy specimens. Specifically, the study aims to develop logistic regression prediction models by training them on microRNA expression data obtained from HNE and OSCC incisional biopsy samples. The primary objective of this research was to develop a robust machine learning model for the accurate identification of oral squamous cell carcinoma (OSCC). Through the utilization of hyperparameter-tuning techniques, our aim was to enhance the model’s performance and refine its analysis of microRNAs derived from standard care diagnostic specimens. Subsequently, we extended the application of this model to 54 specimens of oral potentially malignant disorders (OPMDs) to assess its effectiveness in OPMD risk assessment. The limitation posed by the relatively small dataset utilized in the present study should be acknowledged. This limited sample size presents challenges in generalizing the findings and necessitates cautious interpretation of the results. Nevertheless, clinically, the decision to biopsy or follow up on clinically abnormal lesions is challenging, and our study sought to contribute to early detection and effective management in this context. Additionally, we aimed to evaluate the miRNAs that were identified as most important to the logistic regression model in the process.

## 2. Materials and Methods

### 2.1. Data Acquisition and Ethical Approval

In this study, microRNA expression patterns were investigated through quantitative polymerase chain reaction (qPCR). Two distinct sets of samples were utilized: 30 OSCC and 30 HNE FFPE samples for model development, and a set of 54 unseen OPMD FFPE samples. The OPMD dataset comprised 7 cases of leukoplakia with dysplasia (3 mild, 4 severe), 12 cases of leukoplakia without dysplasia, 13 cases of oral lichenoid lesions, and 22 cases of oral lichen planus. Among the oral lichenoid lesions (OLL), instances were observed to be clinically or histopathologically similar to OLP, but not consistently both. The use of archived FFPE blocks and the entire research protocol received approval from the Human Research Ethics Committee of the University of Melbourne (Ethics ID: 2021-21262-16948-2).

### 2.2. RNA Extraction and Quantification

RNA from 114 FFPE sections were extracted using the Recover All™ Nucleic Acid Isolation Kit by ThermoFisher Scientific. Subsequent RNA quantification using a nanodrop spectrophotometer (ThermoFisher Scientific, Scoresby, Australia) ensured consistency, with precisely 100 ng of RNA utilized for each multiplexed cDNA reaction, according to previously published procedures [9].

### 2.3. cDNA Synthesis and RT-qPCR Analysis

cDNA synthesis reactions were carried out using the Surecycler from Agilent Technologies, implementing a custom multiplex 2-step RT qPCR protocol. MiRXES miRNA select assays facilitated the capture of nuanced expression profiles. Reverse transcription quantitative PCR (RT-qPCR) involved duplicate reactions using the Aria Mx PCR machine (Agilent technology). Cycling conditions included an initial denaturation at 95 °C for 10 min, followed by 40 cycles of denaturation at 95 °C for 15 s and annealing/extension at 60 °C for 30 s. Real-time monitoring through SYBR Green fluorescence measured by the Aria Mx software determined threshold cycles (Ct). miRNAs quantified in this study include miR-21, miR-100, let-7c, miR-24, miR-99, and miR-125b. As controls, a no-template control (NTC) was included to assess for any potential contamination or background signal, and a positive control comprised of fibroepithelial polyp tissue was incorporated to validate the assay performance and ensure the reliability and accuracy of the RT-qPCR results.

### 2.4. Data Analysis and Logistic Regression Model Implementation

Raw data were compiled in Microsoft excel and read into Python 3.8. (Python Software Foundation, https://www.python.org (accessed on 1 October 2022)) as the primary programming language for implementing logistic regression models, leveraging various software packages (Python 3.8) to facilitate data analysis and model development. The core libraries employed for our logistic regression models include scikit-learn, a versatile machine learning library in Python, and NumPy, a fundamental package for scientific computing. The logistic regression models were implemented using the Logistic Regression class from scikit-learn, allowing for seamless integration of logistic regression into our analysis pipeline (Figure 1).

Additionally, we utilized other essential Python libraries, such as Matplotlib for data visualization and creating insightful figures, and pandas for efficient data manipulation and handling. The scientific computing capabilities of NumPy were instrumental in managing numerical data arrays effectively. The code for these implementations can be found on GitHub (https://github.com/npruthiunimelb/Python-Code-) (accessed on 2 April 2024).

The hyperparameter-tuning process was facilitated by scikit-learn’s GridSearchCV and RandomizedSearchCV modules, enabling systematic exploration of model hyperparameters to optimize predictive performance. These tools contributed to the meticulous optimization of regularization strength, penalty type, and solver algorithms for our logistic regression models.

Moreover, Jupiter Notebooks were employed as an interactive computing environment, providing a user-friendly interface for code development, experimentation, and result visualization. 

### 2.5. Data Preprocessing

In the data preprocessing for our logistic regression model, we scaled all numerical features to a uniform scale. This step is essential to bring variables with different units or scales onto a comparable level, preventing any single feature from dominating the model training process. Variables used in logistic regression model were PCR expression levels of 6 microRNAs (miR-21, miR-100, let-7c, miR-24, miR-99, and miR-125b).

### 2.6. Development of Prediction Models

Logistic regression served as the algorithm for risk prediction models. An initial set for model development (30 OSCC, 30 HNE) was divided into training (70%) and test (30%) data. This data was subjected to 10,000 random iterations to ensure robustness during training, optimizing key performance metrics. The decision to use 10,000 iterations in model training strikes a balance between computational efficiency and achieving convergence in logistic regression. This choice ensures stability, allowing the algorithm to explore diverse parameter configurations, adapt to the complexity of the dataset, and improve generalization.

The logistic regression model, developed through a 70:30 split, yielded coefficients for each variable. In machine learning terms, these coefficients represent the learned weights assigned to each feature in the model and indicate the contribution and direction (positive or negative) of each variable’s impact on the predicted outcome. The resulting equation, built from these coefficients, forms the core of the logistic regression model. This equation is then employed by the machine to test the model on diverse datasets, including cases of oral potentially malignant disorders (OPMDs), ensuring a versatile and insightful tool for risk assessment in oral cancer. Key performance metrics, including accuracy, ROC AUC, precision, and recall, were calculated for each iteration. Precision, representing the accuracy of positive predictions, and recall, measuring the model’s ability to capture all positive instances, serve as critical metrics in evaluating the model’s performance. It is noteworthy that the diagnostic categorization of cancer and non-cancer FFPE samples was determined by applying a threshold cutoff of 0.5 for logistic probability values (Table 1).

### 2.7. Hyperparameter Tuning

To optimise model performance, we employed hyperparameter tuning through a grid search approach. This comprehensive tuning process involved fine-tuning key parameters, including the regularization strength (C), penalty type (L1 or L2), and solver algorithms. The overarching goal was to enhance the model’s robustness and reliability through a methodical exploration of hyperparameter space.

Specifically, we conducted two parallel model developments using logistic regression, one with hyperparameter tuning and the other without. For the model with hyperparameter tuning, we systematically varied the regularization strength, penalty type, and solver algorithms across a predefined grid of values. This iterative process spanned 10,000 iterations at each stage, ensuring a thorough exploration of potential configurations.

Concurrently, the initial logistic model, without hyperparameter tuning, served as a baseline for comparison. This model used default settings for regularization strength, penalty type, and solver algorithms. By comparing the performance of these two models on the training data, we aimed to assess the impact of hyperparameter tuning on the logistic regression model’s effectiveness in capturing underlying patterns and relationships within the data. This approach allowed us to ascertain the added value of the fine-tuning process in optimizing the model’s predictive capabilities.

### 2.8. Evaluation of Prediction Models

Finally, we summarized our findings from developing the logistic regression models, to highlight its effectiveness in identifying OSCC. To enhance result interpretation, we used visualizations, like ROC-AUC curves and confusion matrices. These visual aids help provide an understanding of our results. Furthermore, our exploration extended to the examination of logistic regression model coefficients.

### 2.9. Model Extension to OPMDs

After refining the regression model with optimized hyperparameters, we extended its application on the unseen data, which now included the microRNA expression data of 54 OPMD-FFPE samples. These samples were assessed using the parameters established by the trained model, categorizing them into ‘low’ and ‘high’ risk of oral squamous cell carcinoma (OSCC).

## 3. Results

### 3.1. Model Performance on Training Data 

By comparing the initial logistic model and the hyperparameter-tuned model, it became evident that the hyperparameter-tuned model performed better. The hyperparameter-tuned logistic model with 10,000 iterations clearly surpassed the initial logistic model with 10,000 iterations, no tuning in terms of accuracy and ROC AUC, as depicted in Figure 2a,b. The initial logistic model demonstrated an avg. accuracy 0.859 and avg. ROC AUC = 0.864, in comparison to the tuned logistic model with an avg. accuracy = 0.894, avg. ROC AUC = 0.898.

Analysing the confusion matrices of the initial logistic regression model and its hyperparameter-tuned version provided valuable insights, especially when focusing on 18 (30%) randomly chosen samples out of a total of 60 set aside for thorough testing (Figure 3a).

The implementation of hyperparameter tuning in our machine learning model yielded substantial improvements in misclassification rates, signifying the efficacy of optimizing model parameters. Particularly noteworthy was the heightened accuracy in identifying cancer cases, as evidenced by a precision of 0.892 and recall of 0.912. In contrast, the initial model demonstrated lower precision and recall values of 0.853 and 0.885, respectively. These findings underscore the importance of hyperparameter tuning in refining the model’s predictive capabilities, with implications for enhancing its clinical utility in cancer identification, with only one out of the eight actual cancer instances being misclassified as normal (Figure 3b). This improvement highlights the significant impact of hyperparameter tuning in adjusting the logistic regression model, especially in its ability to accurately identify cancer cases—an essential aspect in diagnostic applications.

### 3.2. Deciphering the Significance of Individual miRNAs

Let-7c exhibited the highest positive coefficient (3.155), indicating a robust positive impact on the likelihood of the predicted event. miR-100, with the second-highest positive coefficient (2.188), positively influenced the model’s predictions. Conversely, miR-21, characterized by the lowest negative coefficient (−2.624), displayed a significant negative impact on the predicted outcome. miR-24 (0.651) positively influenced the predicted outcome, miR-99a (−0.552) exhibited a moderate negative impact, and miR-125b (−2.146) had a substantial negative effect, with higher Cq values indicating an increased likelihood of the predicted event. This analysis provides valuable insights into the associations between miRNA expression levels, represented by Cq values, and their contributions to the logistic regression model’s predictions.

### 3.3. Model Performance on Test Data 

The investigation into the performance of the hyperparameter-tuned logistic regression model, initially trained on 30 OSCC and 30 HNE samples obtained from formalin-fixed, paraffin-embedded (FFPE) tissues, revealed a notable enhancement over the untuned counterpart. Building on this, the hyperparameter-tuned model was applied to an unseen test dataset of 54 OPMD- FFPE samples.

The analyses of 54 OPMD FFPE samples demonstrated notable high-risk predictions, particularly in the category of Leukoplakia with dysplasia. Within this subgroup, three cases were identified as high risk, encompassing two instances with mild and severe dysplasia, and one with carcinoma in situ. Furthermore, Leukoplakia without dysplasia exhibited five cases classified as high risk. Similarly, in the groups of oral lichenoid lesions and oral lichen planus, four and nine cases, respectively, were identified as high risk. These findings underscore the model’s efficacy in discerning high-risk lesions, particularly in the context of mild dysplasia, providing valuable insights for targeted monitoring and intervention strategies within the domain of OPMD (Table 2).

It is imperative to acknowledge that the absence of follow-up data precludes the validation of these OPMD predictions against actual long-term outcomes. Nevertheless, this analysis of OPMDs has unveiled compelling insights.

Taken together, the results of our study demonstrated that hyperparameter tuning substantially enhanced the logistic regression model’s accuracy in identifying oral squamous cell carcinoma (OSCC) compared to the initial model. Additionally, the hyperparameter-tuned model shows promise in effectively stratifying the risk of malignancy in oral potentially malignant disorders (OPMDs), underscoring its potential clinical utility.

## 4. Discussion

The investigation presented in this study yielded significant insights into the performance and impact of hyperparameter tuning on a logistic regression model for the identification of OSCC. In comparing the initial logistic model with its hyperparameter-tuned counterpart, it was evident that the latter exhibited superior performance in terms of accuracy and ROC AUC. The study demonstrated substantial improvements in the misclassification rates of OSCC, emphasizing the efficacy of optimizing model parameters. Our study aligns with prior research, affirming the effectiveness of logistic regression in predicting oral squamous cell carcinoma outcomes, reinforcing its significance in the field [21,22]. A recent meta-analysis of healthcare literature found that logistic regression typically performs as well as other machine learning algorithms [23,24].

Analysis of the variable efficiency of the machine learning models provides some considerations for improving predictions in future. In the logistic regression model, three key miRNA variables played a critical role in shaping the predicted outcome. Let-7c demonstrated the highest positive coefficient (3.155), signifying a robust positive influence on the likelihood of the predicted event. Higher cycle quantification (Cq) values for let-7c, indicating lower expression levels, were associated with an augmented likelihood of the predicted outcome. These observations align with the findings of Yap et al. [9]. Likewise, miR-100, boasting the second-highest positive coefficient (2.188), positively contributed to the model’s predictions, and the elevated Cq values for miR-100 enhanced the likelihood of the predicted event. The diminished expression of miR-125b and miR-100 identified by Henson et al. aligns with our study’s emphasis on the role of miRNA dysregulation, particularly let-7c, in refining risk stratification within the oral potentially malignant disorders (OPMD) category [15]. Conversely, miR-21, marked by the lowest negative coefficient (−2.624), exerted a substantial negative impact on the predicted outcome. Elevated Cq values for miR-21, reflecting reduced expression levels, were linked to a diminished likelihood of the predicted event. The meta-analysis findings by Dioguardi et al. highlight the potential of miR-21 as a prognostic biomarker in oral cancer, shedding light on its role in influencing the overall survival outcomes for patients with oral squamous cell carcinoma [25]. This examination provides valuable insights into the intricate connections between miRNA expression levels, as denoted by Cq values, and their roles in influencing the logistic regression model’s predictions. 

The risk stratification within the various OPMD subtypes assessed the model’s effectiveness in identifying high-risk lesions. In the context of the OPMD category, one case diagnosed with mild dysplasia was predicted to be at high risk in the logistic regression model. Potentially these molecular insights, specifically miRNA data, with machine learning techniques may be able to augment traditional histopathological assessments. While conventional histopathology categorized this lesion with mild dysplasia as low risk, these molecular changes might indicate early changes not apparent histopathologically. It may well be that molecular analyses provide a complimentary understanding of changes occurring within tissues and align with recent studies [26] emphasizing the synergistic value of combining machine learning with traditional histopathological methods. 

It is noteworthy that our model presented limitations, as evidenced by two cases of severe dysplasia being diagnosed as low risk. The identification of cases with severe dysplasia being misclassified as low risk by the logistic regression model sheds light on the model’s current limitations. Despite the model’s overall effectiveness, these instances indicate the need for a more thorough understanding of the molecular and histopathological interactions within OPMDs. It is important to note that the OPMD dataset provides cross-sectional information, and we do not possess longitudinal data to determine which cases eventually transformed into malignancy. The absence of long-term follow-up data limited our ability to track the progression of OPMD subtypes over time. 

This discrepancy emphasizes the importance of continuous refinement and updates to the model, incorporating a more extensive dataset and possibly exploring additional molecular markers. Limitations include the model being trained on a relatively small dataset, the absence of longitudinal follow-up data, and the constraints imposed by a limited set of variables. Addressing these limitations is crucial for advancing the model’s accuracy and reliability in predicting the risk of progression in OPMDs. Finally, the significance of long-term follow-up for these lesions is paramount. Tracking the progression or regression of lesions over time provides critical insights into the dynamic nature of OPMD [27]. Future iterations of the model could benefit from a more extensive dataset that includes a diverse range of cases, allowing for a more accurate prediction of risk within OPMDs.

## 5. Conclusions

In conclusion, our study highlights the promising performance of the developed logistic regression model in accurately identifying cases of oral squamous cell carcinoma (OSCC). However, to extend its utility to the broader spectrum of oral potentially malignant disorders (OPMDs), several key considerations must be addressed. The integration of molecular data, using machine learning techniques, incorporating long-term follow-up data, and subjecting the model to external validation emerge as critical steps in refining risk assessment and enhancing predictive outcomes within the OPMD spectrum. This holistic and integrated approach holds immense potential for improving diagnostic accuracy and tailoring personalized management strategies for patients presenting with oral mucosal potentially malignant lesions, particularly in the context of risk prediction. Further refinements and validations are essential for unlocking the full potential of our model in addressing the complexities inherent in OPMDs.

## Figures and Tables

**Figure 1 biomolecules-14-00458-f001:**
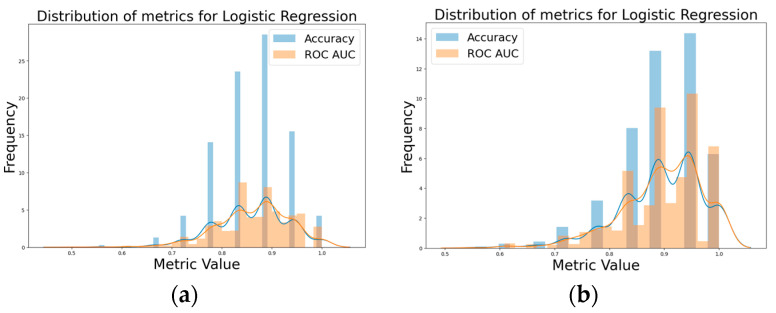
Comparison of performance metrics: (**a**) initial logistic model, (**b**) hyperparameter-tuned logistic model with 10,000 iterations.

**Figure 2 biomolecules-14-00458-f002:**
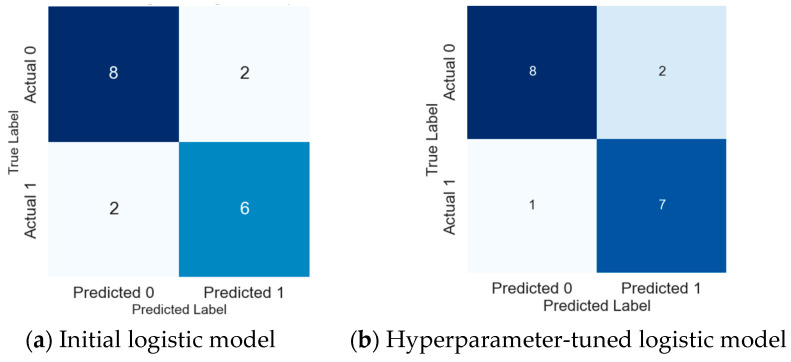
Confusion matrix analysis: (**a**) initial, (**b**) hyperparameter-tuned logistic regression models on 18 randomly selected test samples.

**Figure 3 biomolecules-14-00458-f003:**
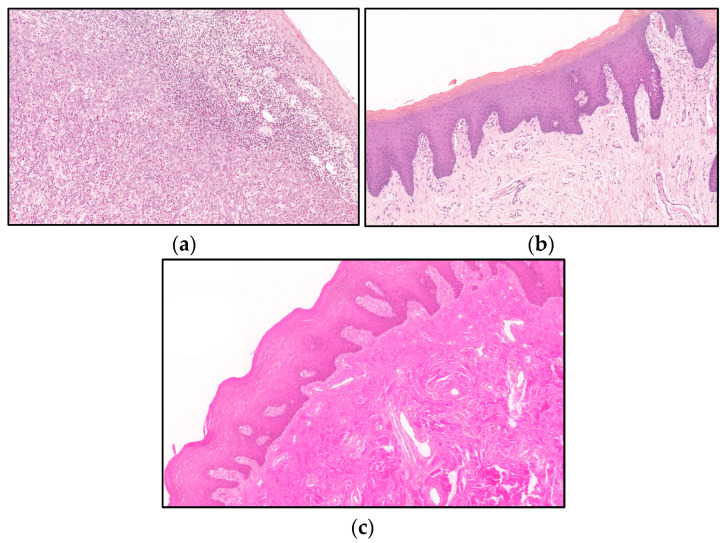
Histopathological images of classification. (**a**) Histopathological image of (**a**) OSCC accurately identified; (**b**) normal tissue sample accurately classified; (**c**) normal tissue misidentified as OSCC (300×).

**Table 1 biomolecules-14-00458-t001:** Methodological workflow summary for developing logistic regression models and extending them to OPMDs.

Workflow	Description
1. Data acquisition and ethical approval.	Obtained FFPE tissue samples for OSCC and HNE (*n* = 30 each) and OPMDs (*n* = 54), ensuring ethical approval.
2. RNA extraction and quantification.	Extract RNA using a commercial kit, standardize input, and quantify RNA concentration.
3. cDNA synthesis and RT-qPCR analysis.	Perform cDNA synthesis and RT-qPCR using custom protocols, including controls for quality assurance.
4. Data analysis and logistic regression model implementation.	Analyze data in Python, implement logistic regression models with hyperparameter tuning.
5. Data preprocessing.	Standardize numerical features and select miRNA markers for logistic regression.
6. Development of prediction models.	Partition dataset, train logistic regression models, and evaluate performance metrics.
7. Hyperparameter tuning.	Fine-tune model parameters using grid search and compare model efficacy with and without tuning.
8. Evaluation of prediction models.	Summarize findings and interpret results using visual aids and analysis of model coefficients.
9. Model extension to OPMDs.	Apply optimized models to OPMD dataset, categorize samples, and incorporate histopathological context.

**Table 2 biomolecules-14-00458-t002:** Hyperparameter-tuned model predictions on unseen dataset across various mucosal abnormalities.

Fifty-Four OPMD FFPE Samples	Number Deemed at Low Risk	Number Deemed at High Risk	Total
Leukoplakia with dysplasia	4 (2 mild dysplasia, 2 severe dysplasia)	3 (1 mild dysplasia, 2 severe dysplasia)	7
Leukoplakia no dysplasia	7	5	12
Oral lichenoid lesions	9	4	13
Oral lichen planus	13	9	22

## Data Availability

Full datasets are available upon reasonable request to the corresponding author.

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
