# Peer review of "Applying Machine Learning for Enhanced MicroRNA Analysis: A Companion Risk Tool for Oral Squamous Cell Carcinoma in Standard Care Incisional Biopsy"

_biomolecules, 2024, doi:10.3390/biom14040458_

Round 1

Reviewer 1 Report

Comments and Suggestions for Authors

Manuscript biomolecules-2891792 proposes a logistic regression model on the identification of oral squamous cell carcinomas (OSCC), utilizing microRNA expression data. 

I couldn’t agree more with the authors that the natural history of OSCC is not fully understood and the effectiveness of OSCC detection in the last years has not improved considerably.

The manuscript is well-structured and written. 

I believe that it is a good effort for a very challenging field.

However, some issues need attention.

In the abstract, the authors claim that the study aims to assess the performance of Machine Learning algorithms in the identification of OSCC patients, but the only algorithm that is being used is the logistic regression model with and without hyperparameter tuning. 

I would suggest using more than one algorithm (e.g., random forest, etc.) and comparing the performance or revising the aims of the study. A statement that a meta-analysis publication proposes the logistic model as the most suitable one, is not good enough.

Also, the “Integrating Machine Learning…” in the title can be changed to “Applying Machine Learning…”

The number of samples (30 OSCC and 30 HNE cases) is limited for ML models. I can understand the difficulty of preparing more samples, but this is something that should be highlighted in the beginning. 

The analysis of 54 OPMD FFPE samples for the validation of the model is a little bit confusing. 

“While follow up data are not available to confirm these predictions, analysis on the previously unseen dataset of OPMD yielded valuable insights, particularly in the context of risk stratification within various mucosal 279 abnormalities associated with OPMD.” It is unclear.

Another thing that would be helpful is the description of the input data (maybe a sample table, or a workflow of the model, etc.), a place where the code is available (e.g., GitHub) and the data, perhaps.

Comments on the Quality of English Language

The manuscript is well-written. It just needs a final proofreading for some minor issues.

Reviewer 2 Report

Comments and Suggestions for Authors

This is a very nice application of ML to OSCC data.

My comments are:

The data set is quite small, even for OSCC. Still, the results are convincing. Do you have the possibility to replicate your results?

In Figure 3, the confusion matrix is indicated. From a pathological point of view, it would be interesting to see the histopathological pictures of samples that were correctly identified by ML (a few examples), and of samples that were incorrectly classified. Altough the ML-classification was based on RNA-data, the histopathology is still the most important diagnostic tool. Therefore, this needs to be incorporated.

Round 2

Reviewer 2 Report

Comments and Suggestions for Authors

Thank you very much for including histopathological pictures.